# Social Representations of Insects as Food: An Explorative-Comparative Study among Millennials and X-Generation Consumers

**DOI:** 10.3390/insects11100656

**Published:** 2020-09-24

**Authors:** Roberto Fasanelli, Ida Galli, Roberta Riverso, Alfonso Piscitelli

**Affiliations:** 1Department of Social Sciences, University of Naples “Federico II”, 80138 Naples, Italy; idagalli@unina.it; 2Department of Political Sciences, University of Naples “Federico II”, 80138 Naples, Italy; roberta.riverso@unina.it; 3Department of Agricultural Sciences, University of Naples “Federico II”, 80055 Portici (NA), Italy

**Keywords:** social representations, cognitive polyphasia, thémata, entomophagy, food culture

## Abstract

**Simple Summary:**

In recent years, a remarkable number of studies have investigated entomophagy from different perspectives. Nevertheless, the theoretical framework of social representations (SRs) has never been used. SRs are organized sets of opinions, knowledge, attitudes, and beliefs about a social object—such as novel foods—co-constructed and shared by a social group. The present research is the first attempt to study entomophagy in the theoretical and methodological framework of SRs. We followed a double trajectory corresponding to two different ways to imagine the contribution of SRs to entomophagy studies. The first one focuses on the role of SRs in the social construction of meanings attached to entomophagy and on their introduction in individuals’ and groups’ thinking frameworks. The second proposes SRs as predictors, even if non-deterministic, of consumers’ behaviors. The results stimulate consideration of social knowledge and cultural variables in food studies using the theoretical framework of SRs.

**Abstract:**

The aim of the research here presented is to describe and compare the social representations of entomophagy co-constructed and circulating among different groups of consumers. Social representations theory (SRT) allows us to understand a social reality that the individual builds based on his own experience in everyday life symbolic exchanges, whose primary function is to adapt concepts and abstract ideas using objectification and anchoring processes. We carried out this research within the structural approach methodological framework. We explored the structure (central core and peripheral schemes) and the content (information, opinions, attitudes, and beliefs) of the social representations of entomophagy by using mixed methodological strategies (hierarchized evocations, validated scales, check-list, projective tool, open-ended questions). Data were processed employing different R packages. The main results show an essential role played by generative processes (objectification and anchoring) as well as cognitive polyphasia and thémata in the co-construction of the social representations of entomophagy. Data could help in understanding the sensory characteristics of “insects as food” that should be used or avoided, for example, in communication aimed to promote entomophagy.

## 1. Introduction

In his preface to Saadi Lalou’s book “Penser manger” [1], Moscovici [2] argues that in the postmodern era our relationship with the table, with eating, that is, with food, has been completely overturned. The author refers to a historical epoch, that of the so-called Baby Boomers and X-Generation [3], which had a relationship with food that can be summarized by Brecht’s maxim *Erst kommt das Fressen*, *dann kommt die Moral*, first comes food, then comes morality. This axiom is opposed by that of the Millennials and Z-Generation [3], for whom, on the contrary, first comes the morality, then comes the food. By subordinating food to a higher ideal of health and conscience (ethics, environment, etc.), these last two generations mentioned have seen numerous taboos emerge and consolidate: anathema about fats, sugars, starchy foods, and so on [4]. Nothing is ruder and more harmful than eating without considering the consequences that excess or forbidden food has on weight, the chemical composition of the body, and even the country’s economy [2]. Nutritional divulgation tries to make us eat austere and dietetic foods and condiments first, which only later are made attractive and delicious by advertising. Today we have put ourselves so well in the hands of experts, continues Moscovici, and we rely so much on doctors and advertisers that we can hardly see the loss of autonomy that comes with it because we no longer know how to eat or sleep “without their permission”. Scientific, as well as social, media continuously report new discoveries. Only a few of these discoveries become known by laypeople and discussed among friends, family, and colleagues. How scientific information taken from the media is integrated into social knowledge shows how people give a common interpretation to the unfamiliar scientific phenomena they encounter in their daily lives. The way people understand, interpret, and describe the scientific discoveries they learn about from the media and from interpersonal communication is influenced by their social representations of the object. This is the case of an important information divulged by Vytenis Andriukaitis, member of the European Union (EU) commission in charge of Health and Food Safety: “Insect producers can make a significant contribution both now and in the future to help us meet the global challenge related to protein, circular economy, and innovation. European policies offer many opportunities to grow and innovate while fully complying with EU food and feed legislation. That is key for the continuous growth and credibility of the sector” [5].

As Materia and Cavallo [6] pointed out, the world population is continuously growing, and with it, the demand for food increases. Processes such as urbanization and globalization are increasingly influencing dietary change for a considerable part of the population. The result is a constant increase in the need for high biological value proteins, the production of which represents a challenge for the future, especially considering that current production techniques (i.e., animal protein farming) not only have a significant environmental impact but also show a low level of efficiency. These techniques produce high levels of carbon dioxide, consume considerable amounts of water, and involve major waste-disposal problems [7]. The European Parliament has indicated that the deficit in protein sources is one of Europe’s most critical problems: the Old Continent imported about 80% of its protein from other countries. Insects can be a sustainable alternative to this problem, for their efficient metabolism and their ability to transform organic waste into high-quality protein [6]. Western countries’ interest in insects as a potential source of food has grown considerably in recent years: the high content of high-quality protein and the sustainability of the production process, compared to traditional sources, primarily meat, have contributed to increasing scientific debate on the topic. The progressive inclusion of insect-based ingredients in the human diet has attracted increasing attention as a valid alternative to overcome the major nutrition challenges the world is facing [8]. However, a diet based on insects (or their components) entails a radical departure from Western societies’ current food traditions. Although recent research shows that consuming insects (raw or processed) provides significant benefits in terms of protein content, social acceptance is, on the contrary, very low in Western societies [9,10,11]. However, insects and their derivatives in food products are not entirely new even in the West: products such as jams and fruit juices contain traces of them, for an estimated average per capita consumption of 250 gr/year [6,12], even if a clear awareness of this is still lacking. Scholars conducted several studies to analyze consumer behavior employing insect-based foods; many of these have identified factors that may positively or negatively influence the degree of acceptance. However, there is a lack of studies aimed to identify social representations of entomophagy and their relationship to food consumption practices. The aim of the study is to explore these representations by means of the lexicon of common sense, laypeople’s dictionaries, the relationships between representations, sensory experiences, and social positioning.

The paper is organized as follows: in Section 1.1, we described the theoretical framework of social representations (SRs) and the importance of the SRs approach to entomophagy. In Section 2, we described the participants and sampling as well as the data analysis performed. Results are illustrated in Section 3. The paper ends by summing up the results and describing possible future developments.

### 1.1. Theoretical Framework

According to Moscovici [13], there is a filter between us and the reality, independent of the sense organs, which has a very particular nature and which reworks information and, in doing so, “creates” the world in which we live and act. This filter is constituted by the social representations that guide us, superimpose themselves on reality, orient individuals, and that can be seen as “practices of knowledge” not subjective, but environmental, cultural, and material, therefore social. They not only have to do with our way of knowing, but they structure, create reality, and orient our behavior socially and culturally, albeit in a way that is neither univocal nor deterministic [14].

Many experimental (see [15] for a review) as well as non-experimental [16,17,18,19,20,21,22,23,24,25,26,27] studies have shown that different representations of the same object can determine different behaviors. As Rouquette and Rateau [28] recall, one does not act following what one thinks (or what one represents), but what one thinks indicates possible action. Therefore, while representations contribute to influencing individuals’ behavior, the link between behavior and representations cannot be explained by a simple theory of information: it is not because people know what they are doing that they necessarily do it. These phenomena can only be understood in the context of social interaction. The way subjects position themselves in the social space, their prior culture, and the process by which the representation has been co-constructed collectively. “In human societies, these elements are more important than ‘truth’ and often even more important than ‘facts’. In order to predict, or even modify, social behaviors, we need to know the representations of individuals as they exist in a given population, and how they function, as well as how they relate to action” [1] (p. 5, our translation). It is the inadequate consideration of such factors that often explains the failure of nutritional reform campaigns. Complex representations allow us to make effective use of our knowledge of the world. To feed us, we need to think. To eat without poisoning ourselves, we need to make complex choices.

As in the case of any other technical-scientific concept, entomophagy is altered when it is spread to laypeople who try to make sense of new, surprising, or unusual information. The theory of social representations (SRT) deals with the processes through which technical-scientific information is integrated into common thought [29], then contributing to the creation of the so-called “common-sense theories”. Nevertheless, the proliferation of the sciences, branches and disciplines, has determined and continues to determine a multiplication of reified/formalized universes. Events, information, theories that dwell in the reified universes must be duplicated, reproduced at a more concrete level, and transferred to those consensual universes (common sense), where it is possible to represent them. To allow this process, two mechanisms are necessary: anchoring and objectification. The former tends to “anchor” unusual ideas, reducing them to daily and family categories and images; the second affords the task of transforming abstract concepts into something concrete, almost physical. Moreover, while the anchoring allows people to transfer unusual concepts in a reference frame where it becomes possible to compare and interpret it, objectification allows the reproduction of this object among visible and tangible things; thus, the object itself becomes manageable. Therefore, we must expect, when studying representations of food, to come across not only a “total social fact” [30] but also profound psychological mechanisms that structure the relationship life of human beings and their culture.

The social psychologist who wants to study reality as it is, and explain behavior, must, paradoxically, disregard “objective reality” and place himself from the outset in the subjective world, lived and acted by individuals. Seen from there, phenomena do not only have material or “cognitive” dimensions; emotions, sensations, and intentions are also relevant [31]. Representational processes actively re-constructs the object; they produce (or recognize) a form in the factory of common sense. For example, think about expressions, increasingly frequent even in a home setting, such as: “My gosh, you made a starred dish!” (The term starred, refers to the evaluation system of an important gastronomic guide, which attributes from one to three stars to evaluate the best restaurants in the world. The raging of the so-called “food-porn” (the over-vision of food in our environment: billboards, bus stops, supermarket posters, TV advertisements, talent shows, and glossy magazines) has made this terminology common sense). This sentence, which seems banal, requires complex cooperation between the actors and prior knowledge of their respective roles. It is only possible thanks to a piece of shared knowledge, the common sense, which is here both in the actors’ minds, and materialized in communication supports. This knowledge and these objects have here made it possible to coordinate the activities of the guests, of the diners, of the cook, who have co-constructed in context a complex sequence of behaviors, and have thus achieved a particular occurrence of what is called a “dinner” and specifically of a “starred” dinner. This sequence seems trivial to us, but would undoubtedly surprise an outgroup participant who would be foreign to our common sense, who would not know the piece in advance like us. Everyday life is a sequence of acts that can only happen in collusion. That is obvious in cases where only a consensus between actors can produce the situation. However, it is also true of everyday acts, such as eating. In order to take part in social life, common sense must be shared. The centrality of this co-constructed and shared knowledge lets us talk about social representations (SRs).The importance of the SRs approach to the study of entomophagy lies in Moscovici’s [32] conviction that society cannot be simplistically reduced to a source of information, but must be considered a source of meaning. People construct questions and search for answers on issues of their interest, rather than merely perceiving and processing information derived from the social context [33]. The daily, permanent presence of food behavior, its capacity to be associated with life experiences in relationships, makes it a privileged support of social relations and cultural acts, as well as its privileged position among peoples’ issues of interest.

### 1.2. Aims and Research Questions

“It is not enough that a food is good to eat, it must also be good to think about it”. This sentence by the famous anthropologist Claude Lévi-Strauss [34] sums up perfectly the situation of today’s consumer-eater who lives in an anxiety-provoking context linked to several phenomena. Some of them are related to our omnivorous nature, others to various characteristics of modern food (distance from the origin of our food, contradictory scientific opinions, sometimes aggressive advertising), and others still to our difficulty in thinking about significant problems of complexity. In addition to this anxiety-provoking context, there are religious food bans, cultural reticence in each country, and ethical considerations. Feeding the planet in the future will, therefore, require that, in addition to the quantitative and qualitative aspects, all consumers should also have the opportunity to find food that is “good to think about”. The aim of the research here presented is to understand if it is “good to think about” insects as food, in other words, which are the social representations of entomophagy co-constructed and circulating in our cultural context.

The framework of social representation as formalized through the structural approach [35,36], assumes that not all the elements of a social representation have the same importance. Some are essential, others less so, others are insignificant. What is essential, if one wants to identify, understand, and modify a representation, is to recognize its organization, the hierarchy of the elements that constitute it, and the relationships they have with each other. Every representation is organized around a central core, which constitutes its fundamental element since it determines its meaning and structure. The central nucleus is a subset of the representation. An element is central when it is qualitatively and not quantitatively relevant. In other words, when it gives meaning to the representation. Around the central nucleus, the peripheral elements are organized. They constitute most of the constituent of a representation and its most accessible dimension. These elements are directly related to the central core and reflect the opinions, attitudes, and beliefs that concern the object. The central core theory [36] has an essential methodological consequence: a social representation study is, first and foremost, looking for its structural elements.

Jean-Claude Abric [37] affirms that we should consider social representations as an organized set of information, opinions, attitudes, and beliefs relating to a given object. Socially produced, they are strongly influenced by the values corresponding to the socio-ideological system and the group’s history that conveyed them, for which they constitute an essential element of the worldview. Even more impressive is the definition and the methodological implication that the author derives from his definition. Being “organized sets” all representations have two components: a content and a structure. In this perspective, to study a social representation means, first of all, to find the principal elements of this structure. The knowledge of the content is not sufficient. The organization of this content gives meaning to the whole representation: two identical contents can correspond to two different symbolic universes and, consequently, subtend two distinct social representations. Many studies have identified gender and age as relevant factors affecting individuals’ willingness to eat insect-based food [11]. For this reason, we hypothesized the existence of different representations of entomophagy in distinct generational categories and between the distinct genders.

Starting from these premises, the present contribution mainly aims at studying the social representation of entomophagy in the structural approach, which means discovering the constitutive elements of its structure (central core and peripheral schemes) and its content (opinions, attitudes, intentions, and generative processes).

The following research questions summarize the core proposals:Do groups of participants belonging to different social categories (gender and generation) produce different entomophagy representations?Which are the more and the less shared contents of the social representation of entomophagy circulating in our socio-cultural context?

## 2. Materials and Methods

### 2.1. Participants

To provide an answer to our research questions, we conducted a fact-finding/exploratory survey on a sample of Italian consumers. The inclusion criterion was represented by having at least heard of the introduction of insects into human nutrition. A convenience sample of 154 participants completely satisfied the inclusion criterion by answering an anonymous web questionnaire’s question. We asked all participants to fill out an online form and recruited using social media (e.g., Facebook, Twitter, WhatsApp chats) and via e-mail lists (e.g., University of Naples student lists). We collected data from May to July 2020 and obtained written informed consent from all participants before participating in the research process. About 58% of respondents were female, and about 42% were male. Only one participant preferred not to declare his gender. The mean age of respondents was 43 years (*SD* = 14.12), and of these, 53% lived in a highly urbanized area, 39% lived in the urban suburbs, and the remaining 8% lived in a rural area. We report the demographic characteristics of the sample in Section A.1., Table A1, Table A2, Table A3, Table A4, Table A5, Table A6, Table A7, Table A8, Table A9 and Table A10.

We divided participants according to age into five distinct generational groups: Z-Generation, Millennial Generation, X-Generation, Baby Boomers Generation, and Greatest Generation [3]. To verify our hypothesis and for reasons of the number of participants belonging to each generational group, we have considered only the two most numerous consecutive generational categories. Specifically, the first group included so-called Millennials (23–39 y.o.) participants (*n* = 60; F = 58.3%, M = 41.7%), from now on MG. The second group, from now on XG, consist of people fitting the so-called X-Generation (40–55 y.o.) (*n* = 57; F = 56.1%, M = 43.9%).

According to guidelines laid down in the Ethics Code of the Italian Association of Psychology, we conducted this study coherently with the Declaration of Helsinki and the European Code of Conduct for Research Integrity [38] (The research protocol was evaluated and approved, as regards methodology and ethical issues, by the Psychological and Social Research Lab “Roberto Gentile”, University of “Naples Federico II” -Research protocol number 0252020).

### 2.2. Data Construction Strategies

In line with the preceding, data collection strategies were articulated and, in some respects, innovative. In particular, to know the internal structure of the social representation of entomophagy, the hierarchical evocations technique was utilized [37]. In the first section of the research tool, we asked each participant to associate the first five nouns that came to his mind from the term inducer “insects as food” and classify them according to importance. Moreover, in the spirit of the use of original tools, better adapted to the nature of the data and in order to avoid the lexical ambiguity that could derive from this type of material, the free associations task was supplemented by five open questions, which prompted the participants to motivate, in writing, the choice of each word. The answers to these questions were useful to understand, through the justifications provided, the meaning, and the attributed sense of each associated term. That is of fundamental importance in the semantic analysis of free evocations [21,25].

The next three sections were devoted to identifying representational contents. In the second section of the questionnaire, we asked participants to answer both semi-structured and structured questions made up starting from the following dimensions: previous knowledge of the object, informative sources, and opinions about it. The third section was devoted to the attitudinal component of the studied social representation by using both the tool validated by Bäckström, Pirttilä-Backman, and Tuorila in 2004 [39] and that recently validated by La Barbera et al. [9]. The 27 statements of the first scale, Social Representation of New Foods Questionnaire (SR_NFQ), were centered on various aspects relating to new foods referable to five dimensions: suspicion of novelties, adherence to technology, adherence to natural food, eating as enjoyment, and eating as a necessity. Items were rated on a 7-point Likert from ‘strongly disagree’ to ‘strongly agree’. The 10 statements of the second scale, Entomophagy Attitude Questionnaire (EAQ), were centered on specific aspects of entomophagy referable to three dimensions: disgust (EAQ-D), interest (EAQ-I), and feeding animals (EAQ-F). Also, in this case, items were rated on a 7-point Likert from ‘strongly disagree’ to ‘strongly agree’. As a supplement, three additional items were administered to measure the intention to eat insects (IDE) in general. This section ended with four items added to ask participants for their willingness to eat specific animals (cow, fish, chicken, and pig) fed with insects (IIE). All answers were collected using a 7-point scale from “strongly unlikely” to “strongly likely”. The fourth section of the research tool was devoted to what we called projective sensory experience (PSE). To identify respondents’ anticipatory gustative sensations regarding “insects as food” and to identify more specific processes of anchoring and its antinomies [40], we used an ad hoc created two-step tool. In particular, (first step) we asked participants to imagine tasting an insect dish and then rate from 1 imperceptible up to 10 very perceptible the following taste-olfactory sensations inspired by the work of Donadini et al. [41]: Sapidity, Bitter tendency, Acidity, Sweet, Spiciness, Aroma, Greasiness-Unctuosity, Succulence, Sweet, Fatness, Persistence. Furthermore (second step), since a representation is always built from a disturbing object (in positive or negative), at the end of the task, we asked our interviewees to indicate, through a specific check-list, which was the most disturbing and least disturbing imagined taste-olfactory sensation. The goal is to know which kind of “sensory anchoring” participants activate in front of an insects dish.

The questionnaire ended by collecting the respondents’ descriptive characteristics (age, gender, job, education, etc.) as well as their eating habits.

### 2.3. Data Analysis Technique

The collected data were initially recorded using spreadsheet software (Microsoft^®^ Excel^®^ 2020) and then used to perform both the analysis of the structural elements of the studied social representation, as well as the statistical analysis of the SRs’ content, operationalized in the following scales: SR_NFQ, EAQ, IDE, IIE, and PSA. All analysis will be presented splitting participants into four sub-groups: MG_F (Millennial females); MG_M (Millennial males); XG_F (X-Generation females); XG_M (X-Generation males)

We first treated the terms evoked by the participants with lexical and categorical analysis. In the lexical phase, we lemmatized free evocations then aggregated it based on the synonymy criterion to obtain clusters of terms substantially coincidental with the manifest meaning [42]. Therefore, using a semantic criterion, terms were further aggregated, starting from their justifications. Each of the obtained clusters was associated with a new label. Every label was identified using, as a selection criterion, the high semantic proximity, and frequency of occurrence of every term aggregated inside of it. Three independent judges completed the whole analytical process. Each judge worked first individually; afterward, they discussed their analysis and agreed on a shared position. We chose an inclusion threshold for the obtained categories, which allowed us to process only those containing words provided by at least 10% of participants. We then processed the obtained data by the software IRaMuTeQ (Toulouse, France) (the R environment’s graphical interface, developed in the Python language) created by Pierre Ratinaud [43]. A prototypical analysis was allowed to reach the elements, which enabled us to hypothesize the central core and the corresponding periphery configuration of the social representation of “insects as food” for each participant group. As Abric [37] underlined, a central element, because of the role that it plays in a social representation, has all the possibilities to be frequent in the verbal expression of its “producers”. Thus, this frequency represents an indicator of centrality if it is completed by more qualitative information, the importance, expressed by the attribution of a hierarchy between elements that the subjects are requested to do. Only the intersection between these two criteria makes the identification of constituent, or significant elements, possible. This method is useful not only to elicit the significant elements but also gives an immediately intelligible output to understand how these elements take place in the organization of the representation. This analysis’s output appears as a “double-entry” table, where elements can be interpreted from the position they have in the four cells. Specifically, the first one (upper left cell, high frequency, and rank) groups the most frequent and essential elements, delimiting the central nucleus area. In the second cell (upper right, high frequency, and low rank), there are the most important peripheral elements (“first periphery” of the nucleus), which give information useful to reconstruct better the social practices related to an SR’s object. In the third one (lower left cell, low frequency, high rank), the elements (of contrast) could configure a nucleus of an SR shared by a minority or be complementary to central elements. The last cell (lower right, low frequency, and low rank) coincides with the area of the “second periphery”, constituted by the elements less present and less important in the structural organization investigated.

In compliance with the entire set of techniques afforded for this type of data, we also treated hierarchized evocations with a similarity analysis [37]. This analysis (a particular type of network analysis) was also supported by the software IRaMuTeQ, which has the advantage of better showing the organizational structure of the significant elements of every social representation. The procedure consists of elaborating a matrix of similarity starting from the selected index, which depends on the relationship’s nature among the considered variables. In our case, we selected the Russel and Rao [44] index. That index is a distance measure, in this case, used because it excluded negative co-occurrences [21]. The output of this analysis consists of a graph on which the social representation’s structural elements are shown with different kinds of link (more or less marked). The selected threshold expresses the relations (and their strength) between structural elements and their network. We elaborated the final graphs using the “maximum tree” logic to provide the best-summarized information about the clustering elements [45]. In the listed figures, the colorful vertices’ size is proportional to the word frequency, and the thickness of the edges indicates Russel and Rao’s index strength of the cognemes link.

Moreover, for SR_NFQ and EAQ scales, we carried out the analysis on the sub-dimensions of the scales (see previous Section 2.2), as the reliability of IDE and IIE items were assessed with Cronbach’s alpha, resulting in 0.94 and 0.97, respectively. Additionally, we performed an analysis of all items of the PSE section. To obtain the individual score for the sub-dimensions of each scale, we calculated the sum of the score of each item belonging to the considered sub-dimensions assessed for each respondent.

After verifying that the data of the item of each scale, as well as its sub-dimensions, were not normally distributed through Shapiro–Wilk tests, a non-parametric version of two-way analysis of variance (ANOVA, Scheirer–Ray–Hare test) was used to show the significance of main effects (gender and age classes/generations of respondents) and their interaction on the social representation’s dimensions (SR_NFQ scale), on attitudes (EAQ scale), on intentions (IDE-IIE items), as well as on projective sensory variables (PSE scale). Pair-wise multiple comparisons, used as a post hoc test, were not performed because both the gender factor and the age/generation classes factor have only two levels.

Statistical analyses were carried out in the “R” environment [46]. Applying the method implemented in IRaMuTeQ, we obtained the prototypical and similitude analysis. A non-parametric version of two-way ANOVA (Scheirer–Ray–Hare test) was carried out by the “scheirerRayHare” function in “rcompanion” package. If not specified, *p*-value < 0.05 was considered statistically significant.

## 3. Results

### 3.1. Representations’ Structure

#### 3.1.1. Prototypical Analysis

As already mentioned, these results were obtained by analyzing the intersection between the rank of importance (IR) and frequency (Freq) of appearance, utilizing the IRaMuTeQ software.

The analyses of the structural elements (Table 1) in the left upper quadrant—which constitutes the hypothetical central core of the Millennial females’ representation of “insect as food”—reveals that the category Disgust (which aggregates terms like disgust, repugnance, vomiting, nausea, etc.) reaches the highest number of preferences (Freq. 23; 65.7% of these interviewees) and the most important position (Importance Rank- IR 2.0). Many participants write that: “I feel like I am going to puke at the thought of swallowing them…” (MG147_F) (Participant alphanumeric identifier).

In the same quadrant, we found Insects_species, which aggregates all the references to every kind of insects and invertebrates (locust, crickets, ants, flies, cockroaches, larvae, scorpions, etc.). This category reaches fewer preferences (Freq. 11; 31.4%) but a better position of importance (IR 1.9). In this case, participants justify their choice using short sentences: “I remember an oriental video that shows this as food” (MG111_F). The third constitutive element of the possible nucleus is Dirty/Impure, which has nine co-occurrences and is an element shared by the 25.7% of Millennial females. This category incorporates statements like: “Insects are usually found in places that are not necessarily clean, hygienic” (MG33_F). The right upper quadrant identifies the first periphery of the social representation of “insects as food” constituted by only one category. The Millennial females involved in the research frequently refer to the consistency on the palate of insects, as shown in the next sentence classified, like all the similar, as Crunchy (Freq. 12; 34.3%; IR 4.0): “Insects have a thin outer shell” (MG28_F). In the lower right quadrant, we found the second periphery of the studied social representation, characterized by the structural elements entering the common sense theory on “insects as food” or that are becoming less important, then leaving it. In other words, here we can find all the most fluctuant cognitions about the object. The only category in this quadrant Insect_parts (Freq. 15; 42.9%; IR 3.7) not connected to the sensation previously described, but to the personal relationship with insects and their legs, wings, and antennas, was testified to by the interviewee who wrote, “Because they are the part of the insects that makes me the most uncomfortable” (MG11_F). Categories in the lower left quadrant might be explained by the presence of a sub-group of participants who share a different vision of the object. These women experience Fear (Freq. 4; 11.4%; IR 2.2) because they “are afraid of all insects” (MG106_F) and they think that these creatures are Infectious (Freq. 4; 11.4%; IR 1.8) even if, thankfully, they are eaten mainly in Other_Countries (Freq. 3; 8.6%; IR 2.3).

Table 2 shows the results for Millennial males. In this case, the first quadrant of the prototypical analysis output is different compared to the results obtained from the females of the same generation.

The first category, Disgust (Freq. 17; 68.0%; IR 2.6), contains the largest number of free associations in this sub-group which underpin a negative representation of the “insects as food”, as indicated by many respondents in their naïve theories: “A part of the population neither consumes nor habitually nor at most rarely insects. That is because they are rated as living creatures of the lowest category. Hence the sense of psychological repulsion”. (MG60_M). Even if non-expert, these people made a large number of references to Proteins (Freq. 12; 48.0%; IR 2.8) with a great accuracy “Proteins derived from insects are alternative sources of protein, at low cost and with less impact on the environment for their production, compared to others of animal origin” (MG74_M). Free association justifications such as “Insects as food make me think of Eastern, or South American cultures” (MG50_M), are the reason because we labeled Other_Countries (Freq. 8; 32.0%; IR 3.2) the category in the first periphery of the internal structure of this social representation.

In the lower right quadrant, there is the category Insect_species (Freq. 7; 28.0%; IR 2.9) that was constructed to summarize the freely associated nouns which identified specific insects and invertebrates. The label Crunchy (Freq. 4; 16.0%; IR 4.0) comes from the most frequent word used by these participants to explain the sensation provoked by being “forced” to imagine themselves eating insects: “I have been thinking about skewers made of fried insects” (MG43_M). Inside the category Resource (Freq. 3; 12.0%; IR 4.3) and Fear (Freq. 3; 12.0%; IR 3.7) were classified expressions like “I believe that food resources will become increasingly scarce in the future and insect consumption will also be used to cope with this scarcity” (MG57_M), even if “The idea of ingesting something that disgusts me scares me” (MG98_M).

The lower left cell includes the last two categories, complementary, in reverse order, of those present in the nucleus. The first is Future (Freq. 6; 24.0%; IR 2.2) “because, for a sustainable world, I believe that in the future we will have to eat insects” (MG41_M). The second is Dirty/Impure (Freq. 5; 20.0%; IR 2.4) “because drinking water gives you a clean feeling, and eating bugs gives me a dirty feeling” (MG74_M).

Table 3 shows the results for X-Generation females. In the first quadrant, devoted to central elements, we find three categories. The first one, Disgust (Freq. 16; 50.0%; IR 1.9), summarizes a series of statements not significantly different from that of participant XG21_F, who wrote: “I find it disgusting to associate food with insects”. The second one, Dirty/Impure (Freq. 10; 31.3%; IR 2.1), refers to all those social positions that consider “insects as food” a feed “dirty, and impurity-filled” (XG107_F). The third cogneme belonging to the hypothetical central core of this representation coincides with the most frequent noun freely associated by the X-Generations females to denote the nutritional value of this food: Proteins (Freq. 7; 21.9%; IR 2.7).

The first periphery of this structure, generally influenced by respondents’ social practices, is characterized by Crunchy (Freq. 7; 21.9%; IR 3.3) and Fried (Freq. 5; 15.6%; IR 4.4). Some justifications to the elicited evocations clarify their collocation in this cell: “Crunch is the noise that comes from stepping on them or chewing them as they are” (XG146_F); “I made an association with fried seafood” (XG143_F). The second periphery is essentially linked to respondents’ social identity and intergroup relations. Two of the three categories contained in this quadrant Other_Countries (Freq. 4; 12.5%; IR 4.0) and Poverty (Freq. 3; 9.4%; IR 3.0) clearly testify to these processes especially looking at interviewees’ explanations. X-Generation females point out that we do not eat insects in our country because “Asia is the continent where people usually eat insects” (XG149_F) and also because “usually in poor countries they eat insects” (XG35_F). The last category belonging to this cell collects all the references to Insects_species (Freq. 3; 9.4%; IR 3.0). Also, in this case, the left lower cell is characterized by a core complement element. Coherently with the central nucleus dominated by the Disgust, some interviewees belonging this group totally Refuse (Freq. 3; 9.4%; IR 2.3) the idea of “insects as food” because they “cannot take that into account” (XG162_F).

Table 4 synthesizes the results for X-Generation males. The figurative nucleus of these participants’ social representation of the entomophagy is clearly dominated by the Disgust (Freq. 10; 43.5%; IR 2.1) for every Insects_species (Freq. 8; 34.8%; IR 1.9)

The upper right cell shows the closest element to the nucleus, Dirty/Impure (Freq. 11; 47.8%; IR 2.8), identified by affirmations such as: “There are bugs that alight on everything” (XG92_M). The most distant elements to the core of this internal structure are listed in the lower right quadrant. Here we find Poverty (Freq. 4; 17.4%; IR 3.5), which confirms the idea, present only among participants belonging this generation, that “Insects are eaten in poor countries” (XG10_M). A strategy of distancing the problem, which is also confirmed by references to Culture (Freq. 3; 13.0%; IR 3.7) since members of this group have argued that: “Insects are diametrically opposed to the idea of food in our culture” (XG136_M). Anyhow, in this cell, we also find references to Crunchy (Freq. 4; 17.4%; IR 4.2) justified by the fact that people “think about the possible chewing noise” (XG88_M). The last cell of the table, lower left quadrant, characterized by low frequency but high importance, contains ambivalent information. The consciousness of the nutritional value of insects, testified by the presence here of Proteins (Freq. 4; 17.4%; IR 2.2), is supported by references to the Future (Freq. 3; 13.0%; IR 2.3) but also contradicted by the re-affirmation of Refuse (Freq. 3; 13.0%; IR 3.2) as underlined by the respondent MG7_M, who informed us about the fact that he is “never going to taste insects voluntarily”.

From the classification obtained through prototypical analysis, it is possible to note, as expected, the evident salience of the concept of disgust in the social representation of “insects as food”. It was most frequently mentioned at a high level of importance by both women and men, older and younger, participants. That is confirmed by the centrality of this element in the configurations presented in the following figures, through which it will be possible to reconstruct the existing interrelations between the constitutive elements of the structure of the explored social representation. In fact, as mentioned [37], it is not enough to know the content dimension of a social representation. It is the organization of this content that gives meaning to the entire representation. It also permits the comparison between groups: identical contents which may correspond to a different symbolic universe and, consequently, imply different social representations.

#### 3.1.2. Similarity Analysis

The first configuration concerns Millennial females’ in-depth conceptions of the entomophagy. As already mentioned, we chose the Russell and Rao [44] coefficient (RR) to weigh the semantic link’s strength between each structural component of the representation.

As shown in the graph, the most important category is Disgust. Due to 6 links (Figure 1) that add a RR coefficient of 0.9 (76.9% of the whole graph coefficient), this concept is the element with the highest degree of relatedness. This category’s centrality in the social representation structure is of evidence thanks to the interconnections existing between the distinct justifications that respondents use to explain their free associations. For example, the student MG12_F argues: “I think they are crunchy, but once you eat them, they are disgusting” (Disgust-Crunchy = RR 0.29). In each parenthesis, we reported the links between the semantic categories implied in quotations, complemented by the relative index of Russel and Rao (RR), to indicate their strength.

Similarly, respondent MG3_F wrote that “eating insects is disgusting because being insects they can neither be clean nor hygienic” (Disgust-Dirty/Impure = RR 0.23). The multifaceted nature of the anticipated sensation of disgust related to entomophagy is also confirmed by the interviewee MG76_F, who claims that “it is disgusting to eat insects, because (I think) they can cause infections”. This cluster of elements, which de facto identifies the core of these participants’ representation, is also characterized by antithetical conceptions. On the one hand, we have the link between Disgust and Fear (RR 0.11), consistent with what we described so far. See, for example, the explanation given by the interviewee MG36_F, who claims: “I feel a sense of disgust just at the thought of eating insects, because I am afraid of all insects”. On the other hand, we also find a particular curiosity towards insects as food, as testified by some participants’ statements when they assert: “I am battling between curiosity and disgust” (Disgust-Curiosity = RR 0.09).

The remaining two clusters of concepts present in this graph account for both a nutritional (Crunchy-Proteins = RR 0.09) and culinary (Crunchy-Insects_parts-Fried = RR 0.06) dimension of the representation of insects as food, as well as a globalized dimension (Insect_species-Proteins-Other_Countries = RR 0.06), influenced by media communication. The following quotations support this interpretation: “I have seen many photographs, videos, TV shows and documentaries about Asian markets where insects were sold as food” (MG14_F); “In some documentaries, I saw that frying is one of the most frequent insect preparation techniques (for example in Thailand)” (MG167_F).

The graph correspondent to male participants belonging to this sub-group (Figure 2) shows a less articulated internal structure than females. In fact, in this case, we have only two different clusters that are very dissimilar in weight. The most crucial aggregate, also, in this case, is the one composed by the closest cognemes to the category labeled Disgust (5 arches that together add a RR coefficient of 0.96–72.7% of the whole graph coefficient). The second aggregate generated starting from the category Proteins, is composed of 4 links that add a RR coefficient of 0.36 (27.3% of the whole graph coefficient). Also, in this situation, when we asked participants to justify their free associations, they used brief expressions strictly linked between them that help us to understand the described semantic architecture. That is the case of the respondent MG75_M, who wrote: “In my opinion, many people may find the inclusion of insects in a diet disgusting. I believe that insects are already consumed in Asia” (Disgust-Other_Countries = RR 0.24). Disgust is provoked by every kind of insect (and non-insects, i.e., scorpions) for these respondents as underlined by participant MG24_M who assert: “Disgust is the first feeling that came to mind because the cockroach is the first insect that came to my mind” (Disgust-Insects_species = RR 0.16). Although they reject the idea of “insects as food”, the Millennial males recognize this resource’s importance for the coming years. The associative chain of the subject MG41_M is emblematic in this sense: “The idea of eating them is pretty repulsive to me. For a sustainable world, I believe that in the future, we will have to eat them. To save the world could be an option” (Disgust-Future = RR 0.16). Finally, the connection between Disgust and Fear (RR 0.12) is well summarized by the participant MG98_M who affirm: “The idea of ingesting something that disgusts me scares me”. The second network generates from the node Proteins is linked to Resource with a very weak arch (RR 0.12). Participant MG41_M explains the nature of this connection: “Insects are a very rich source of protein, fundamental nutrient, low cost and low impact on the environment for their production”. The statement of the respondent MG49_M, for its part, helps us to understand the reason for the link between Proteins and Dirty/Impure (RR 0.12): “I see the unclean insect as a vector of disease even though it is a source of proteins”. To obtain the protein it contains, an insect must be eaten. That seems to be a non-trivial obstacle for some participants, such as MG134_M, who argues that “most insects have a crunchy skeleton, but because they walk and fly everywhere, ‘they do not care’ about hygiene” (Dirty/Impure-Crunchy = RR 0.12).

Exactly like the one just described, the internal structure of the social representation of the females belonging to the X-Generation subgroup has a two-dimensional configuration (Figure 3). Again, the central core is characterized by the constellation of elements that gravitate around Disgust (4 arches that together add a RR coefficient of 0.58–57.4% of the whole graph coefficient). The second cluster begins from the category Proteins and contains three nodes and two arches that together add a RR coefficient of 0.27 (26.7% of the whole graph coefficient). The graph’s strongest link is that between Disgust and Dirty/Impure, which presents a RR coefficient equal to 0.23. The last category mentioned generates two peripheral arcs towards the Crunchy and Insect_species nodes with the same RR coefficient: 0.08. Respondents’ statements are again necessary to understand these interconnections. Participant XG55_F, for example, wrote: “Just the thought of eating insects makes me horrified and disgusted, because they make me think of dirty environments”. Moreover, interviewee XG55_F asserts: “I thought about the crackling of fried grasshoppers, but it disgusts me […] I think their smell is neither pleasant nor healthy”. Finally, the X-Generation woman identified by the code XG103_F declares: “I find it filthy to associate insects and food […] I think of grasshoppers as possible food insects”. The figurative core of this representation, generated by Disgust, keeps Poverty (RR 0.12) and Refuse (RR 0.08) within it. The first, because “In poor countries they eat insects” (XG35_F). The second, because “the thought of it (inductor: insects as food) makes me horrified, so I refuse the idea”. (XG55_F). In the second cluster, participant XG8_F helps us to understand the trajectory from Proteins to Fried (RR 0.15): “Fried crickets are appetizing […] I have tasted cricket flour […] insects are high protein food”. Likewise, interviewee XG139_F highlighted the reason for the one from Fried to Other_Countries (RR 0.12): “Usually there (evocation: Asia) they eat insects […] Frying covers the flavors”.

By contrast with all the previous configurations, the representational structure of X-Generation males presents two main categories, Disgust and Dirty/Impure, although the first one is the one with a higher connectedness because of its four links (Figure 4) that add a RR coefficient of 0.48 (49.5% of the whole graph coefficient). Nevertheless, the link between these two categories, the strongest in the graph (RR 0.26), is fascinating to understand these people’s naïve theory of entomophagy. Participant XG121_M, for example, helps us to understand: “Eating insects makes me vomit […] I feel repulsion for eating insects […] I associate eating insects with dirt”.

Eating any insect causes a strong sense of repulsion in these interviewees, mostly because they imagine the noise that chewing it would cause. Interviewees XG23_M, XG38_M, and XG44_M they gave us only a part of the long and imaginative list of insects freely associated to the inductor: “cockroaches, ants, spiders, spiders, larvae, grasshoppers, silkworms, scorpions, bees, mosquitoes, wasps, flies” (Disgust-Insects_species = RR 0.09). Subject XG88_M clarifies the relationship between repulsion and ‘snacking effect’: “I think I would find them disgusting […] I thought about the possible noise caused by masticating it” (Disgust-Crunchy = RR 0.04).

Like the X-Generation females, males also seem to think that there is a relationship between entomophagy and poverty and even more clearly. For these participants, the link is between dirt and misery, as the interviewee XG10_M remembers: “Personally they disgust me, they are so dirty […] I would vomit at the mere thought of eating them […] Insects are eaten in poor countries” (Dirty/Impure-Poverty = RR 0.13).

The two central categories generate two peripheral antithetical schemes. The first that starts from Disgust highlights the existence of a subgroup of people belonging to this social category, which considers, like the subject XG57_M, that. “It is a novelty that we will slowly have to get used to for the consumption of proteins that will be increasingly scarce […] I believe that food resources in the future will be increasingly scarce and to cope with this scarcity, and we will also turn to the consumption of insects” (Future-Proteins = RR 0.09). The second, which branches off from Dirty/Impure, identify interviewees as XG158_M, who reject even the idea of feeding on insects, relegating these practices to other cultures too different from their own: “I reject the idea a priori because just to see them disgusts me […] because in my culture it is absurd” (Refuse-Culture = RR 0.09).

### 3.2. Representations’ Content

#### 3.2.1. Opinions and Attitudes

As the previous section of this work has been dedicated to analyzing the internal structure of the social representations of entomophagy, the present one will be dedicated to the description of their contents, operationalized in opinions and attitudes.

Stoetzel, Palmonari, Cavazza, and Rubini [47] define opinion as an evaluative statement on a controversial issue that presents characteristics of instability, plasticity, and specificity. According to them, we choose to “evaluate” respondents’ opinions operationalizing it, employing closed-ended questions, as well as IDE and IIE items. The attitude instead (here “measured” by using SR_NFQ and EAQ scales), as stated by Moscovici [29], is a critical organization that expresses an evaluative orientation towards an object. This orientation can be revealed through a global behavior or a series of answers with an ordinary meaning. Considered in relation to attitudes, opinions are simple and manifest answers, while attitudes are organized and latent answers. Below is the mutual articulation of these essential components of the social representations of “insects as food” elaborated by the research participants.

Opinions, generally the most consensual part of a social representation, show a total convergence among those who belong to the four different subgroups of participants. Table A11, Table A12, Table A13, Table A14, Table A15 and Table A16 in Appendix A summarize the results obtained about this representational dimension, from which it is possible to deduce the following: 70.1% of the entire sample believes that the insects used as food come from specialized farms, while only 23.1% are convinced that they come from the natural environment. However, 6.8% of respondents believe that both channels of supply are plausible. Moreover, 71.8% believe that insects as food can be eaten only after undergoing processing and transformation, while 24.8% are convinced that they can also be ingested without any treatment. Also, this time the number of those who consider both solutions valid is very low (3.4%). Among those who believe that preparatory processing for ingestion is necessary, 58% believe that this should only affect some species of insect, while the remaining 42% are convinced that all insects must be processed before being put on the table. Among the items dedicated to opinions and beliefs, there were also two dedicated to the perception of risk related to entomophagy. It is interesting to note that 55% of Millennials believe the risk of contracting a disease by ingesting insects is moderately improbable to extremely improbable, while 18.3% of the sub-sample think it is moderately to extremely probable. The members of the X-Generation subgroup, on the other hand, are 43.9% convinced that the risk of contracting diseases by eating insects can vary between moderately improbable and extremely improbable. In comparison, a more consistent 33.3% think it is between moderately probable and extremely probable. Asked how serious the health risks of using insects as food could be, the majority (55%) of Millennials responded that they considered them little or not at all serious. Only 21.7% of these respondents considered the possible health risks related to entomophagy to be moderately to extremely serious. Once again, the members of the X-Generation sub-sample seem to be a little more pessimistic. Only 38.6% believe that the risks to human health deriving from insects’ consumption vary from slightly serious to not at all serious, while a consistent 36.8% consider them moderately or extremely serious.

The participants’ cognitive-affective orientations are summarized in Table 5. The interviewees’ attitude towards food novelties in general (SR_NFQ scale) is negative and shared by all participants regardless of gender and age. Looking at the nuances between the median values of the four sub-samples, we can observe the following. X-Generation subjects are generally suspicious of food novelties compared to Millennials, who, to a lesser extent, show a slight openness to such novelties.

Both on this attitudinal sub-dimension and the one related to the adherence to new technologies, the participants seem to be substantially ambivalent, considering that the median values close to the value 4 indicate that the first half of the respondents were in disagreement or undecided with the assertions submitted to their judgment. Much sharper appears to be the positions related to the further issues investigated. In particular, both concerning the predilection for natural foods and living food as enjoyment, all respondents seem to be on the attitudinal continuum’s positive pole. On the contrary, the low median values, made more stable by the low interquartile range values, related to the subdimension Food as a necessity, indicate that the participants in the research care about what they eat, even if hungry, and therefore they care how their food is produced, as well as the fact that they need much information on new foods. The attitude towards entomophagy (EAQ scale) is also negative across generations and genders. The participants show great disgust towards insects as food and, consequently, little interest in tasting insect-based dishes under any circumstances (restaurants, parties, etc.). Also, concerning insects used as food for other edible animals, the participants in this research show that they do not have an exact position, as evidenced by the median values for each sub-sample for the Feeding animals sub-dimension of the scale in question. Moreover, these values make unequivocal the interviewees’ position about their intentions to directly taste insects as food: it is extremely improbable. More likely, on the contrary, they seem to be referring to the indirect entomophagy on the IIE scale, on which males of the Millennials generation record a median of 5 to indicate moderately possible that they can eat meat or fish bred with insect feed. To check for the interaction of gender with the respondents’ age classes, a non-parametric two-way ANOVA (Scheirer–Ray–Hare test) was performed on each dimension of the SR’s scale, on each dimension of the EAQ scale, and on intentions IDE and IIE. The age classes of respondents considered are Millennials and X-Generation. For all dimensions of the SR scale, EAQ scale, IDE and IIE intentions there is no statistically significant interaction effect between gender and generation, while we observed the main effect “Age classes” to be statistically significant for SR_NFQ-1 (H = 4.21; *p* = 0.040), SR_NFQ-2 (H = 4.74; *p* = 0.029), SR_NFQ-4 (H = 13.87; *p* = 0.000), and the main effect “Gender” to be statistically significant for EAQ-D (H = 6.91; *p* = 0.008), EAQ-F (H = 5.53; *p* = 0.019), IDE (H = 5.06; *p* = 0.024) and IIE (H = 7.97; *p* = 0.005). (Appendix A—Table A17)

#### 3.2.2. Projective Sensory Experience

As previously explained (see Section 2), to better identify processes of anchoring and objectification specific for a “gustatory object”, we created an ad hoc tool, the PSE. The purpose was to know which sensory characteristics of “insects as food” should be avoided and which one must be promoted to increase the willingness to eat them. Table 6 shows the median value of the projective sensory variables for each sub-group of interviewees.

In order to check for the interaction of gender with the age classes of respondents, a non-parametric two-way ANOVA (Scheirer–Ray–Hare test) was performed on each projective sensory variable. The age classes of respondents considered are Millennials and X-Generation.

For all sensory projection variables, there was no statistically significant interaction effect between gender and age classes. In contrast, the main effect “Gender” was statistically significant (H = 5.39; *p* = 0.020) only for the Spiciness variable and the main effect “Age/Generation classes” was significant (H = 4.89; *p* = 0.027) only for the Greasiness-Unctuosity variable (Appendix A—Table A18).

In general, the most perceived sensation, even if imagined, among Millennials is the Bitter tendency, whereas among X-Generation’s participants is Persistence. The second perceived sensation into the Millennials group is Greasiness-unctuosity, while into X-Generation, one is the Bitter tendency. Persistence is the third choice for Millennials as well as Sapidity for X-Generation. Acidity, Spiciness, and Aroma cover the next position among Millennials and the last two sensations among X-Generation interviewees. In the fifth position, there are Sapidity for Millennials and Acidity and Greasiness-unctuosity for X-Generation. Considering the less perceived sensations Sweet and Fatness are at sixth position among Millennials, although the first of these sensations have the same rank among X-Generation’s people. The Sweet tendency occupies the seventh position for both the groups of participants. The eighth position is occupied by Succulence for Millennials and by Fatness for X-Generation. Finally, the ninth position is covered by the X-Generation’s interviewees’ absolute least perceived sensation. Imagining tasting insects as food, they do not feel they are characterized by Succulence.

Asked about the imagined taste-olfactory sensation that most disturbed them, participants answered as reported in the contingency Table A19, Table A20, Table A21 and Table A22 in the Appendix A. In the columns, there is the gender of respondents, in the rows the taste-olfactory sensations chosen. Data in the column profiles represent the relative frequencies (expressed as percentages) of the different taste-olfactory sensation within each respondent’s gender category. In each generational category considered, to highlight gender differences concerning taste-olfactory sensations, we can compare the column profiles with the respective average profiles. The column profile of a specific sensation that differs positively from the respective average profile will characterize that gender category. In other words, if the column profiles of both male and female respondents who reported a specific imagined taste-olfactory sensation as their most disturbing are closest to their respective average profiles, then males and females do not differ for that specific characteristic.

In Table A19, the column profile of the females reporting Greasiness-Unctuosity as their most disturbing imagined taste-olfactory sensation is far from their respective average profile, so Greasiness-Unctuosity characterizes millennial females and differentiates them from millennial males. The column profiles of male respondents who reported both Bitter tendency and Persistence as their most disturbing imagined taste-olfactory sensations are far from their respective average profiles, so these specific sensations characterize millennial males and differentiate them from Millennial females. The column profiles of both male and female respondents who were reporting Succulence as their most disturbing imagined taste-olfactory sensation are closest to their respective average profiles and, therefore, they do not differentiate males from females.

In Table A20, the column profile of the females reporting Sweet tendency as their least disturbing imagined taste-olfactory sensation is far from their respective average profile, so Sweet tendency characterize millennial females and differentiate them from millennial males. The column profile of the males reporting Spiciness as their least disturbing imagined taste-olfactory sensation is far from their respective average profile, so Spiciness characterizes millennial males and differentiates them from millennial females. The column profiles of both male and female respondents who reported both Sapidity and Sweet as their least disturbing imagined taste-olfactory sensations are closest to their respective average profiles and, therefore, they do not differentiate males from females.

In Table A21, the column profile of the X-Generation females reporting Bitter tendency as their most disturbing imagined taste-olfactory sensation is far from their respective average profile, so the Bitter tendency characterizes females and differentiates them from males. While the column profiles of X-Generation male respondents who reported both Acidity and Greasiness-unctuosity as well as Persistence as their most disturbing imagined taste-olfactory sensations are far from their respective average profiles, so these specific sensations characterize males and differentiate them from females. The column profiles of both male and female respondents who were reporting Succulence as their most disturbing imagined taste-olfactory sensation are closest to their respective average profiles, and, therefore, do not differentiate males from females.

In Table A22, the column profile of the X-Generation females reporting Sapidity as their least disturbing imagined taste-olfactory sensation is far from their respective average profile, so Sapidity characterizes females and differentiates them from males. The column profiles of X-Generation male respondents who reported both Spiciness and Sweet as their least disturbing imagined taste-olfactory sensations are far from their respective average profiles, so these specific sensations characterize males and differentiate them from females.

## 4. Discussion

Agreeing with Levi-Strauss [34], we chose foods not as good to eat but good to think about. Therefore, we can ask ourselves: what is good to think about what we eat? What if we eat insects?

This paper was entirely devoted to trying to answer these questions attempting to understand what people think when they think of eating insects. In other words, which naïve theories they co-construct thinking about insects as food; definitively, which social representations of entomophagy are circulating in our social context.

It is worth noting that our hypothesis to find different social representations in the four subgroups of participants involved in the study was not verified. It seems to us we encountered such a particular typology of social representation that Moscovici [32] defined as agonal. He distinguishes among: closed/hegemonic SRs, characterized by representational elements uniformly distributed and shared throughout the whole population; agonal/critical/polemical SRs, characterized by representational elements approximately similar in the entire population, but with meanings determined by differing and even contrasting values; open/emancipated SRs, characterized by representational elements distributed among the various subgroups of a population so that it is necessary to bring them together to find out their coherence.

In our case, representational subdimensions are comparable in the entire sample, but with meanings defined by some divergent beliefs, elements visible only at a deeper level of qualitative, hermeneutical observation of the data. All representational structures gravitate around the same central core Disgust. Nevertheless, peripheral elements, the ones influenced by respondents’ social practices, gave crucial additional information to understand the real meaning every group attribute to representation. For Millennial females, the imagined sensation of something Crunchy in the mouth plays an important role. Effectively, the exoskeleton of insects has a significant influence on the texture. Insects are crunchy, and sounds accompanying their eating resemble the sound of crackers. Some of these participants, to the preliminary question that allowed access to filling in the questionnaire (related to the knowledge of the object of the representation “insects as food”), replied that they had even tasted it, although not making frequent use of it. Males of the same generational group, on the contrary, attributed higher importance to the nutritional properties of “insects as food”, balancing the sensation of Disgust with the idea of the source of Proteins. The same connection resulted important also for X-Generation females. Nevertheless, these women and much more men belonging this generation assigned a central role to the belief that insects, even if edible, are Dirty/Impure. According to previous studies (see [11] more or less all the described representational structures are embedded by the idea of insects as dirty and dangerous. This argument about the issue of cleanliness and its link to the concept of purity, according to Rochira [48] call upon the notion of ‘basic thémata’ introduced by Ivana Markovà [49] (p. 213) who considered that interrelation the lintel which maintains “physical distancing from disgusting people, protection and preservation of the social order”. Thémata is a notion initially proposed by Holton [50] and reworked by Moscovici and Vignaux [51]. The thémata constitute a set of general conceptions, of force-ideas, deeply anchored in a group’s collective memory. They imply dyads or triads of oppositional notions, making it possible to explain the formation of real strands of thought and the emergence of certain social controversies. They seem to have a generative and normative power in forming a representation, shaping the new information on those already existing. In Moscovici’s opinion [14] thémata are “bits” of knowledge or shared beliefs. Clear/dirty and pure/impure belongs to this particular symbolic categories deeply rooted in collective imagination that, as pointed out by Speltini and Passini [52], give rise to intense emotional reactions of either attraction or repulsion towards impure and dirty persons/groups, thus legitimizing prejudice, discrimination, and social exclusion. So, even if laypeople’s entire discourse on entomophagy is full of antinomies, suspicion/trust, natural/artificial, enjoyment/necessity, it seems that dirty/clean and pure/impure play the most generative role in the social construction of the studied representations. In this way, we, too, could explain the profound nature of disgust through contamination. Participants in this research tend to have a stereotyped and undifferentiated perception of insects as impure. Partially because of intergroup discrimination, mostly because, according to Verneau et al. [11] (p. 31) also in this case “the association of some insects with feces and decaying matter could have led to psychological contamination of all insects, making the entire category disgusting”.

That is a destiny that common sense does not attribute only to insects. As reminded by Debucquet et al. [53] historians analyzed the importance of the beliefs associated with red meat considered disgusting, when bloody, because it was “still alive”. Other research on the perception of food germs has shown that blood in red meat is still nowadays perceived as risky because of the survival of the belief in “spontaneous generation” among laypeople. Similarly, many non-consumers are disgusted by oysters because of their appearance as something alive and viscous. Some scholars [53,54] identified these two physical characteristics, alive and viscous, as the most common and most potent sources of disgust. Many consumers easily eat fresh, live oysters thanks to their culture, which considers suitable these specific sensory characteristics and, above all, by classifying them as luxury and rare food. Ingesting oysters, or escargot or caviar, for example, means eating positive symbols, desirable “worlds”, which strongly reduce the sense of disgust. In other words, the social representations associated with some “disgusting” food by consumers identifying in it classy, sophisticated, and healthy food, “helps” its consumption [55,56,57]. The time has probably come to work for “insects as food” to become the escargot, caviar, and oysters of the 21st century. The first step could be to dismantle those fundamental representational dimensions generated by cultural anchors and objectification in the other by itself, i.e., in the Orientals and the poor.

Our interviewees found a clear strategy to cope with the “entomophagy risk” by placing it somewhere else, far away from us, in Other_Countries such as India, China, etc. A strategy of distancing the problem which is also confirmed by frequent and transversal references to culture.

In the case of opinions and beliefs about insects as food, participants in the research show a precise application of “cognitive polyphasia” [13]. Different, and sometimes competing, bits of knowledge are used to give sense to the social reality. These different and frequently contradictory ways of thinking, meaning, and practices are argued to co-exist because they feature situated knowledge. That is, each mode of knowledge is linked to the context of its production. Thus, inconsistencies are accommodated, as each representation is argued to be locally consistent [58,59,60,61]. For this reason, the same people who adopted coping strategies such as collocating entomophagy in other countries and cultures, classifying it as a poor eating habit, and so on, show specific technical competences regarding insect farming and production strategies that allow their transformation into food. Despite this reified knowledge, they recur in parallel to consensual explanations, showing strong prejudices about insects’ healthiness as food. Therefore, an oscillation is typical of that specific representational process called cognitive polyphasia that, among our participants, we also saw in the swinging between scientific knowledge linked to insects as the protein bank of the future and objectifications in dust, slime, dirt, feces, etc. Controversial issues give rise to polemical social representations.

The attitudinal component of the studied representation confirms this convergent heterodoxy. In the chosen framework, attitude does not merely identify a set of particular and heteronomous opinions or answers, but a mechanism capable of actively imposing order on these opinions and answers. In other words, it performs a regulatory function, allowing subjects to make a selection within their cognitive and emotional universe. The behavior is not its most immediate concrete manifestation, even if, when it manifests itself, its content and value are exact. The predictivity of behavior from an attitude, anyhow, is a very controversial issue, which also finds some theorists of social representations dissenting (see [62] for a review). They—and we are among them—consider attitudes as one of the components of a representation and believe that the genesis of behavior has a probabilistic, multi-determined, and virtual status within the SRT. Moving back to the research results, it is clear that the attitude toward entomophagy is overall negative, but some intergroup differences are worth noting. Millennials showed a slight openness to food novelties, even envisaging the possibility of accepting indirect entomophagy.

According to Kouřimská and Adámková [63], sensory properties are significant criteria accompanying the consumption of edible insects. The taste and flavor of insects are very different and affected by pheromones occurring at the insect’s surface that could be washed off by rinsing. Chitin and exoskeleton also influence the organoleptic composition, but cooking insects gives them the taste of added ingredients, i.e., sugar, salt, or spices. Cooking and conservation methods also affect the color of edible insects. All this information, nevertheless, is not already common sense. For this reason, one of the aims of this study was to identify the generative process of anchoring, in this case, a sensory-taste anchoring, to understand which kind of anticipatory sensations people use to take its decision of eating or avoiding “insects as food”. In some way, we obliged participants to activate this process by stimulating their imagination and, to the best of our knowledge, it is something that has never happened in the study of social representations, always studied as products, less as processes [64].

Results, in this case, offer an exciting perspective. Succulence is the most disturbing and harmful sensation shared by all Millennials. Females belonging to this subgroup are specifically disturbed by the imagined Greasiness-Unctuosity and males do not tolerate the Bitter tendency and Persistence in the taste of the imagined insect dish. Among this generation of interviewees, Sapidity and Sweet are the less disturbing shared sensations. However, the Sweet tendency characterized the projective reactions of millennial females and Spiciness the male ones. Concerning X-Generation’s participants, it is interesting to note that for these people, also, Succulence is the most disturbing and damaging sensation shared. In this subgroup, however, Bitter tendency characterizes females’ imagined worst sensation and Acidity, Greasiness-Unctuosity, and Persistence the male ones. No shared sensation resulted in this subgroup where females better tolerate Sapidity’ and males Spiciness and Sweet of the imagined dish. These data could help who are searching for the sensory characteristics of “insects as food” that should be avoided and which ones should be used, for example, as marketing strategies to promote entomophagy. In general, we are convinced that it is necessary to take entomophagic discourses out of their status of “esoteric curiosity” to make them an object of study with its own cultural specificity that needs to be much more stressed.

## 5. Conclusions

What we commonly call “eating behavior” summarizes, in synthetic terms, an immense field of passions and knowledge, rules, and signs that occupy our imagination. Anthropologists have been more vigorously interested in the subject, especially in the so-called primitive cultures, more rarely the sociologists, while social psychology of socio-constructivist orientation has remained in a refuge. The difficulty is not in noticing this refuge, but in explaining its persistence. When we see the extent of industrial and economic investment and, even more so, the place that everything related to cooking, food occupies in daily conversation, in the publication of books, and mass communication, we wonder why the psychology of social knowledge has retreated from it, or perhaps put it at a distance. Nevertheless, there is no other field of social life in which the values, ideas, or words of a human group manifest themselves, with greater resonance, most intimately and directly, than that of food. We hope that the originality of the current research can reside, first of all, in its theme: drawing the outlines of the socio-cultural psychology of eating behavior called “entomophagy”. That was its project, and, whatever the judgment on the method or concepts used, it remains its ambition. Combining a socio-constructivist perspective on entomophagy and a functionalist approach to sensations (even if imagined) may provide a comprehensive and promising theoretical framework for the study of the role of immaterial imaging in the study of social representations, in particular of anchoring process, involving relevant consequences for the research methodologies and instruments used to investigate these phenomena.

### Limitations and Future Developments

This study has limitations, indeed. The most important is perhaps the convenient nature of the sample and its relatively small size. The research process here proposed could also be applied across a wider geographic area to understand the social representations of entomophagy throughout Italy and further identify regional and contextual influences. Comparisons between North–South, city–province, and center–periphery might be fruitfully explored by future research. Future work could also address whether residential area [65] influences the relationships between interviewees and insects as food. Finally, with the aim of developing a model of the problem [66], it might be useful to introduce additional indicators, considering not only subjective perceptions and self-reported evaluations [67,68], but also objective environmental data [69,70], in order to achieve a more comprehensive understanding of the timely topic addressed in the paper.

## Figures and Tables

**Figure 1 insects-11-00656-f001:**
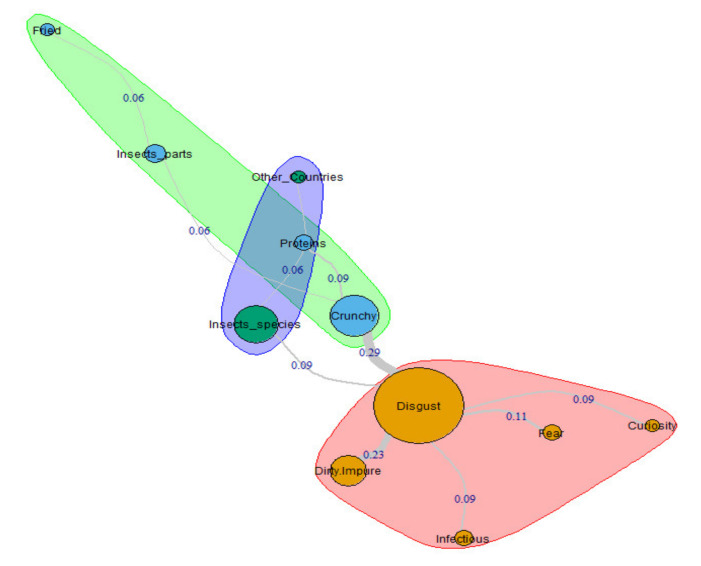
Internal configuration of the Millennial females’ social representation (SR).

**Figure 2 insects-11-00656-f002:**
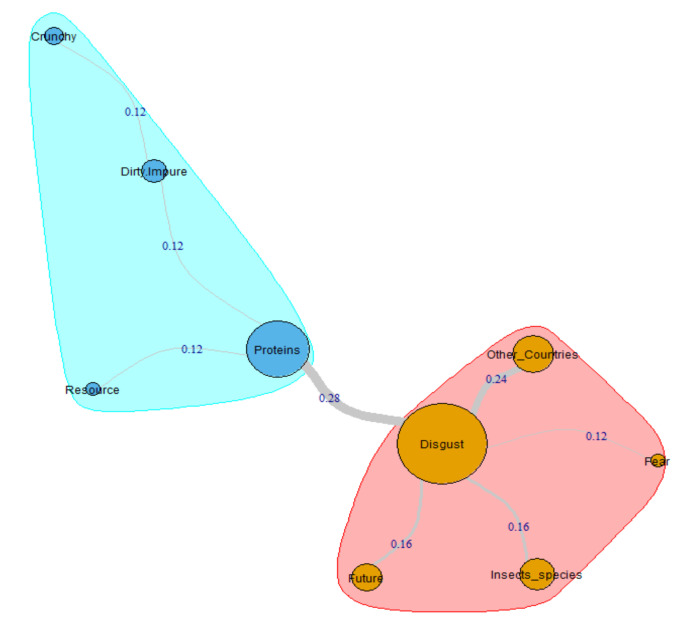
Internal configuration of the Millennial males’ SR.

**Figure 3 insects-11-00656-f003:**
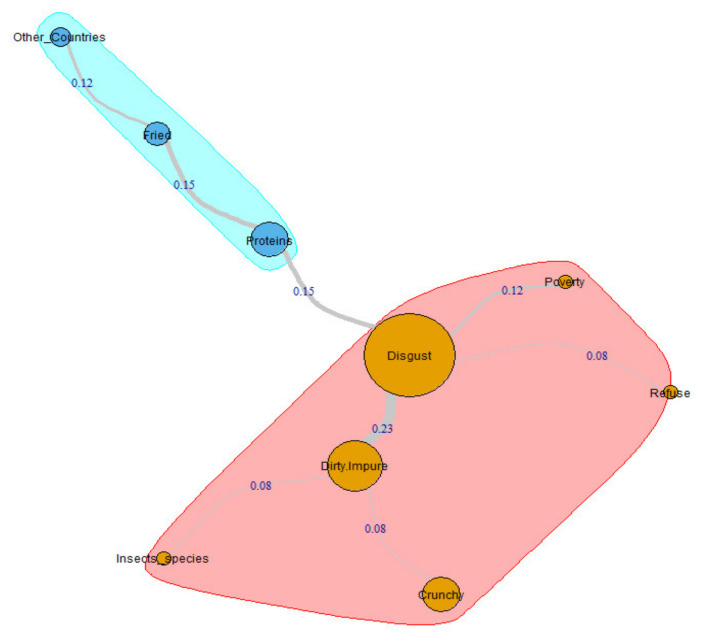
Internal configuration of the X-Generation females’ SR.

**Figure 4 insects-11-00656-f004:**
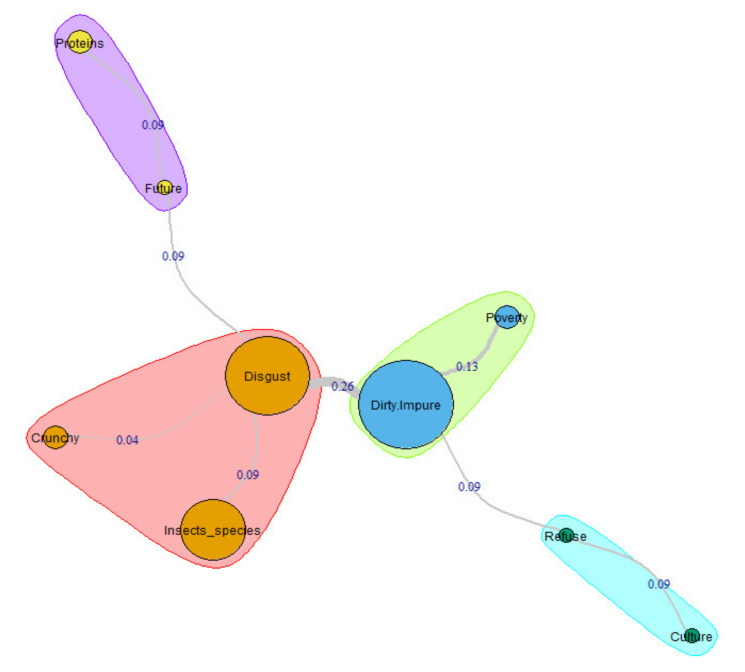
Internal configuration of the X-Generation males’ SR.

**Table 1 insects-11-00656-t001:** Millennial females—prototypical analysis.

Frequency/Importance	Importance Rank ≤ 2.58		Importance Rank > 2.58	
**FREQ. ≥ 7.36**	Disgust	23–2.0	Crunchy	12–4.0
	Insects_species	11–1.9		
	Dirty/Impure	9–2.4		
**FREQ. < 7.36**	Fear	4–2.2	Insects_parts	5–2.8
	Infectious	4–1.8		
	Other_Countries	3–2.3		

**Table 2 insects-11-00656-t002:** Millennial males—prototypical analysis.

Frequency/Importance	Importance Rank ≤ 2.83		Importance Rank > 2.83	
**FREQ. ≥ 7.22**	Disgust	17–2.4	Other_Countries	8–3.2
	Proteins	12–2.8		
**FREQ. < 7.22**	Future	6–2.2	Insects_species	7–2.9
			Crunchy	4–4.0
	Dirty/Impure	5–2.4	Resource	3–4.3
			Far	3–3.7

**Table 3 insects-11-00656-t003:** X-Generation females—prototypical analysis.

Frequency/Importance	Importance Rank ≤ 2.79		Importance Rank > 2.79	
**FREQ. ≥ 4.86**	Disgust	16–1.9	Crunchy	7–3.3
	Dirty/Impure	10–2.1	Fried	5–4.4
	Proteins	7–2.7		
**FREQ. < 4.86**	Refuse	3–2.3	Other_Countries	4–4.0
			Poverty	3–3.0
			Insects_species	3–3.0

**Table 4 insects-11-00656-t004:** X-Generation males—prototypical analysis.

Frequency/Importance	Importance Rank ≤ 2.62		Importance Rank > 2.62	
**FREQ. ≥ 5.56**	Disgust	10–2.1	Dirty/Impure	11–2.8
	Insects_species	8–1.9		
**FREQ. < 5.56**	Proteins	4–2.2	Poverty	4–3.5
	Refuse	3–2.0	Crunchy	4–4.2
	Future	3–2.3	Culture	3–3.7

**Table 5 insects-11-00656-t005:** Median values and interquartile range of Social Representation of New Foods Questionnaire (SR_NFQ), Entomophagy Attitude Questionnaire (EAQ) and Intentions scales.

Scales		*Millenials Generation*		*X-Generation*	
		Females	Males	Females	Males
SR_NFQ	1. Suspicion of novelties	4.5 (2)	4 (2.5)	5 (1.875)	4.5 (2.25)
	2. Adherence to technology	4 (2.5)	4.5 (1.25)	4 (2.25)	3.5 (1.75)
	3. Adherence to natural food	6 (2)	5 (1)	6 (1)	6 (1)
	4. Food as enjoyment	6 (2)	6 (1)	5.25 (1.75)	5 (2)
	5. Food as a necessity	2 (1)	2 (1)	2 (1.5)	2 (0.75)
EAQ	Disgust	6 (3)	5 (2)	6 (3)	5 (2.5)
	Interest	3 (3)	4 (4)	2 (4)	4 (4)
	Feeding animals	4 (3.5)	4.5 (1.5)	4 (1.875)	4 (3)
Intentions	IDE	1 (2)	2 (4)	1 (1.75)	2 (3.5)
	IIE	4 (3)	5 (2)	3 (4)	4 (3.75)

**Table 6 insects-11-00656-t006:** Median value of the projective sensory variables across groups.

Sensations	*Millenials Generation*		*X-Generation*	
	Female	Male	Female	Male
*Sapidity*	6	5	6	6
*Bitter tendency*	7	8	7	7
*Acidity*	6	6	5	4
*Sweet*	4	4	4	4
*Spiciness*	5	7	5	6
*Aroma*	5	7	6	5
*Greasiness-unctuosity*	6	8	5	4
*Succulence*	2	2	1	3
*Sweet tendency*	4	3	3	4
*Fatness*	3	5	3	3
*Persistence*	6	7	8	7

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
