# Peer review of "Social Representations of Insects as Food: An Explorative-Comparative Study among Millennials and X-Generation Consumers"

_insects, 2020, doi:10.3390/insects11100656_

Round 1
Reviewer 1 Report
Very good manuscript, very original theme on human food and the temptation or disgust of insects, well worked from an empirical and theoretical point of view. Errors in the bibliography should be corrected. The anthropological dimension of the study of social representations, which goes beyond the research for their core and periphery, should also be emphasized.
The paper deals with the investigation of social representations of entomophagy.
The paper is very well anchored to the field of social psychology and the theory of social representations from a theoretical and methodological point of view. In my knowledge, is the first paper dealing with entomophagy in social psychology.
It is clear and easy to read.
Author Response
Dear Reviewers,
first of all, we want to thank you for your suggestions. Constructive feedback from colleagues is something precious to an author. We hope we have been able to take your suggestions to improve our work.
Below is a detailed description of the changes we have made.
Crossed fingers, we are now asking to revise the new manuscript: Social representations of insects as food: An explorative-comparative study among Millennials and X-Generation consumers.
Best regards.
Yours faithfully,
Roberto Fasanelli, Ida Galli, Roberta Riverso and Alfonso Piscitelli.

Reviewer 2 Report
The scientific contribution is real, especially in the field of social representations. The theoretical framework used is relevant and heuristic. The data processing meets the expected standards in social psychology of representations. The data construction strategy is interesting by its combination of squantitative and qualitative data.
The manuscript may be a bit long, some parts could be reduced without affecting the quality of the article.
The question of gender could be developed with regard to statistical processing carried out according to sex.
An error is present in table9 (frequency of "insert_parts"?).
Author Response

(The authors gave the same response as above.)
